# Variant-specific symptoms of COVID-19 in a study of 1,542,510 adults in England

Matthew Whitaker [1,2,10], Joshua Elliott[3,4,10], Barbara Bodinier [1,2], Wendy Barclay [4], Helen Ward [3,5,6], Graham Cooke [3,4,6], Christl A. Donnelly [1,5,7], Marc Chadeau-Hyam [1,2,11] & Paul Elliott [1,2,3,6,8,9,11] ✉

Infection with SARS-CoV-2 virus is associated with a wide range of symptoms. The REal-time Assessment of Community Transmission −1 (REACT-1) study monitored the spread and clinical manifestation of SARS-CoV-2 among random samples of the population in England from 1 May 2020 to 31 March 2022. We show changing symptom profiles associated with the different variants over that period, with lower reporting of loss of sense of smell or taste for Omicron compared to previous variants, and higher reporting of cold-like and influenza-like symptoms, controlling for vaccination status. Contrary to the perception that recent variants have become successively milder, Omicron BA.2 was associated with reporting more symptoms, with greater disruption to daily activities, than BA.1. With restrictions lifted and routine testing limited in many countries, monitoring the changing symptom profiles associated with SARS-CoV-2 infection and effects on daily activities will become increasingly important.

A meta-analysis of studies from the first wave of the pandemic identified 30 symptoms reported in multiple studies[1], including common influenza-like symptoms (cough, fever, myalgia/fatigue, headache, sputum production), and less common but more specific symptoms including change or loss of sense of smell or taste.

Previous community-based studies have assessed the degree to which symptom data can predict polymerase chain reaction (PCR) positivity for SARS-CoV-2, and have used variable selection and ranking techniques to identify the most important (set of) symptoms for case identification[2–4]. Further studies have indicated that symptom profiles may differ between variants of SARS-CoV-2[5–7].

The relationship between symptom profile and cycle threshold (Ct) value from PCR testing (an established proxy for viral load[8–10], which in turn correlates with infectiousness[11,12]) has yet to be fully investigated. Identifying individuals who are more likely to be (i) infected, and (ii) infectious on the basis of symptom profile would have

clinical value as governments move away from mass testing programmes and mandatory isolation measures.

Here, we use regression modelling and variable selection models in the large community-based REal-time Assessment of Community Transmission −1 (REACT-1) study that was in the field approximately monthly from 1 May 2020 to 31 March 2022 to i) describe the symptom profiles of the main variants of SARS-CoV-2 that have been dominant in England and worldwide over this period, namely wild-type, Alpha, Delta and Omicron BA.1 and BA.2, and ii) identify the symptoms that are most predictive of high viral load, and hence infectiousness, for each variant.

## Results

### Descriptive and univariable analysis

The characteristics of our study population are summarised in Fig. 1 and Supplementary Tables 1 and 2. It comprised 1,542,510 adults aged

[1]School of Public Health, Imperial College London, London, UK. [2]MRC Centre for Environment and Health, Imperial College London, London, UK. [3]Imperial College Healthcare NHS Trust, London, UK. [4]Department of Infectious Disease, Imperial College London, London, UK. [5]MRC Centre for Global Infectious Disease Analysis and Jameel Institute, Imperial College London, London, UK. [6]National Institute for Health Research Imperial Biomedical Research Centre, London, UK. [7]Department of Statistics, University of Oxford, Oxford, UK. [8]Health Data Research (HDR) UK London at Imperial College, London, UK. [9]UK Dementia Research Institute at Imperial College, London, UK. [10]These authors contributed equally: Matthew Whitaker, Joshua Elliott. [11]These authors jointly supervised this work: Marc Chadeau-Hyam, Paul Elliott. ✉e-mail: p.elliott@imperial.ac.uk

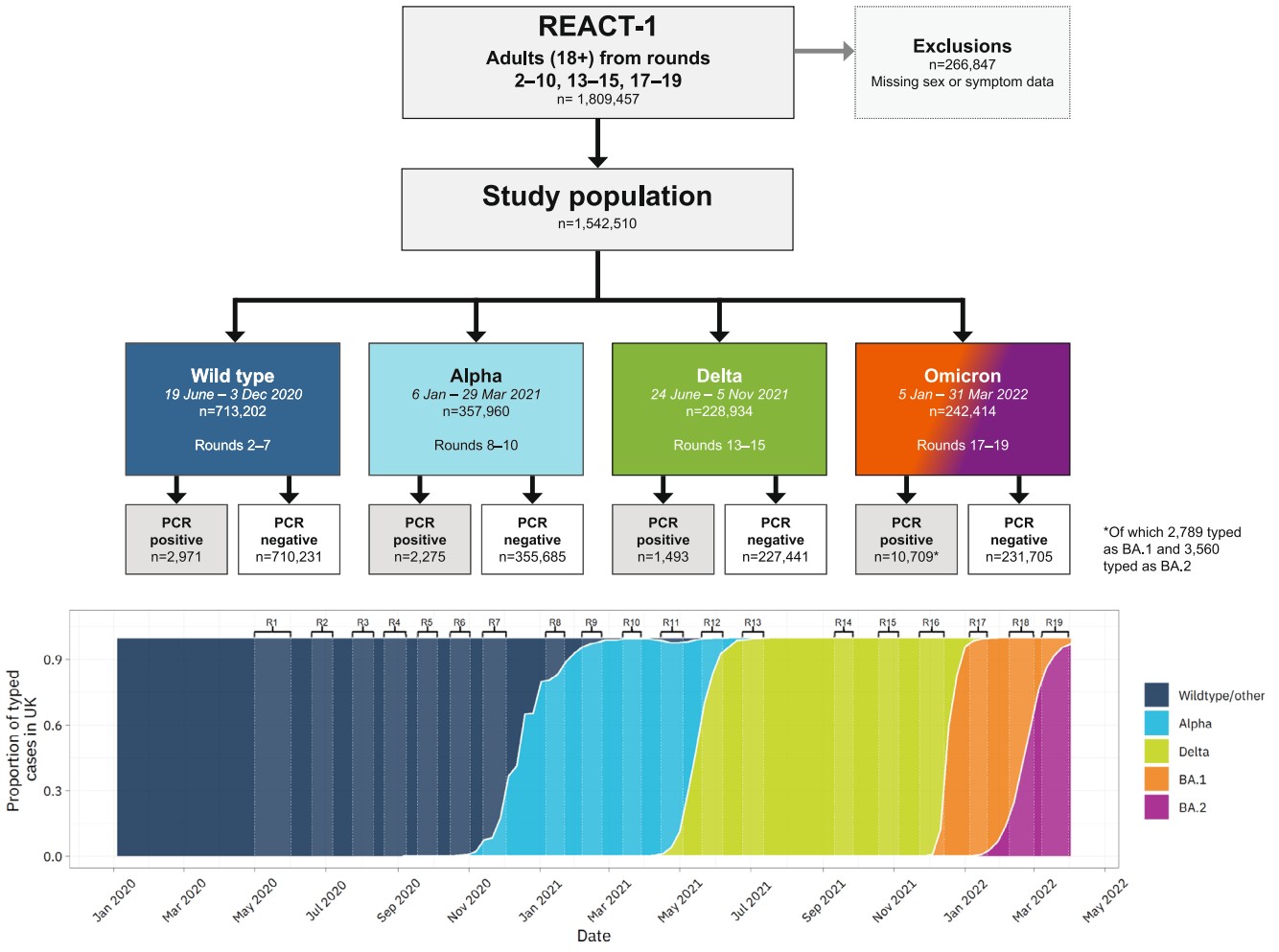

**Fig. 1 | Study population flow-chart.** Variant prevalence data in bottom panel is from GISAID[26].

18 and over, including a total of 17,448 swab positive individuals: 2971 (0.4%, 95% Confidence Interval (CI) [0.4,0.4] unweighted prevalence) for wild type; 2275 (0.6% [0.6,0.7]) for Alpha; 1493 (0.7% [0.6,0.7]) for Delta and 10,709 (4.4% [4.3,4.5]) for Omicron variants (Supplementary Table 1).

The proportion of swab positive individuals reporting any of 26 symptoms (symptoms listed in Supplementary Table 1) was highest in those infected with BA.2 (75.9% [74.4,77.2], compared with 70.0% [68.3,71.6] in those with BA.1, 63.8% [61.3,66.2] in those with Delta, 54.7%, [52.7,56.8] in those with Alpha and 45.0% [43.3,46.8] in those with wild-type) (Table S2). Background prevalence of symptoms was also highest during January–March 2022, when Omicron dominated: 21.9%, [21.7,22.0] of all respondents reported one or more symptoms, compared with 13.5% [13.4,13.5] during the wild-type period (Supplementary Table 1).

Those infected with BA.2 reported an average of 6.0 (95% CI 5.8,6.2) symptoms in the week prior to PCR testing, compared with 2.70 (2.6,2.8), 3.4 (3.2,3.6), 4.6 (4.4,4.9) and 4.6 (4.5,4.8) for wild-type, Alpha, Delta and BA.1 respectively (Supplementary Table 2). A larger proportion of people with BA.2 reported that their symptoms had affected their ability to carry out day-to-day activities 'a lot' (17.6% [16.3,18.8]) compared with those infected with BA.1 (10.7% [9.6,11.9]) or Delta (10.5%, [9.1,12.2]) (Supplementary Table 2).

All symptoms were positively associated with swab positivity for all variants (Fig. 2, Table S3). The odds ratio for swab positivity of 'any' vs 'none' of 26 symptoms was highest for BA.2 (OR = 12.9 [11.9,14.0], compared with 5.7 [4.8,5.6], 6.0 [5.1,7.1], 9.5 [8.6,10.6] and 9.6 [8.8,10.5]

for wild-type, Alpha, Delta and BA.1, respectively) (Supplementary Table 3, Fig. 2).

Unlike for wild-type, Alpha, and Delta, where the highest odds ratios for swab positivity were for loss or change of sense of smell (ORs 49.7 [44.3,55.7], 37.8 [28.6,50.0] and 73.4 [64.2,83.9], respectively) or taste (ORs 35.9 [31.9,40.4], 38.9 [29.9,50.6] and 68.1 [59.4,78.0] respectively), for BA.1 and BA.2 influenza-like and cold-like symptoms were relatively more predictive of swab positivity, and loss or change of sense of smell or taste relatively less so. Within BA.1 and BA.2, the highest odds ratio of all symptoms was for fever: ORs were 18.4 [16.5,20.5] for BA.1 and 30.2 [27.7,33.0] for BA.2, compared with 12.9 [11.1,15.1] and 17.2 [15.1,19.5] respectively for loss or change of sense of smell and 16.0 [13.9,18.5] and 21.3 [18.9,24.0] respectively for loss or change of sense of taste (Fig. 2, Supplementary Table 3). In a sensitivity analysis, further adjusting for time since symptom onset attenuated the odds ratios, but the patterns across variants remained consistent with the main analysis (Supplementary Fig. 7).

A pooled analysis (Methods) reinforced the findings from the univariable analysis after adjusting for SARS-CoV-2 prevalence and background symptom prevalence, showing that Alpha and Delta were associated with increased symptom-specific odds ratios across most symptoms, while Omicron BA.1 was associated with lower odds ratios across most symptoms, and especially for the loss of sense of smell or taste (Supplementary Fig. 2). Omicron BA.2 was associated with increased odds ratios vs BA.1, most notably for cold-like symptoms and chills.

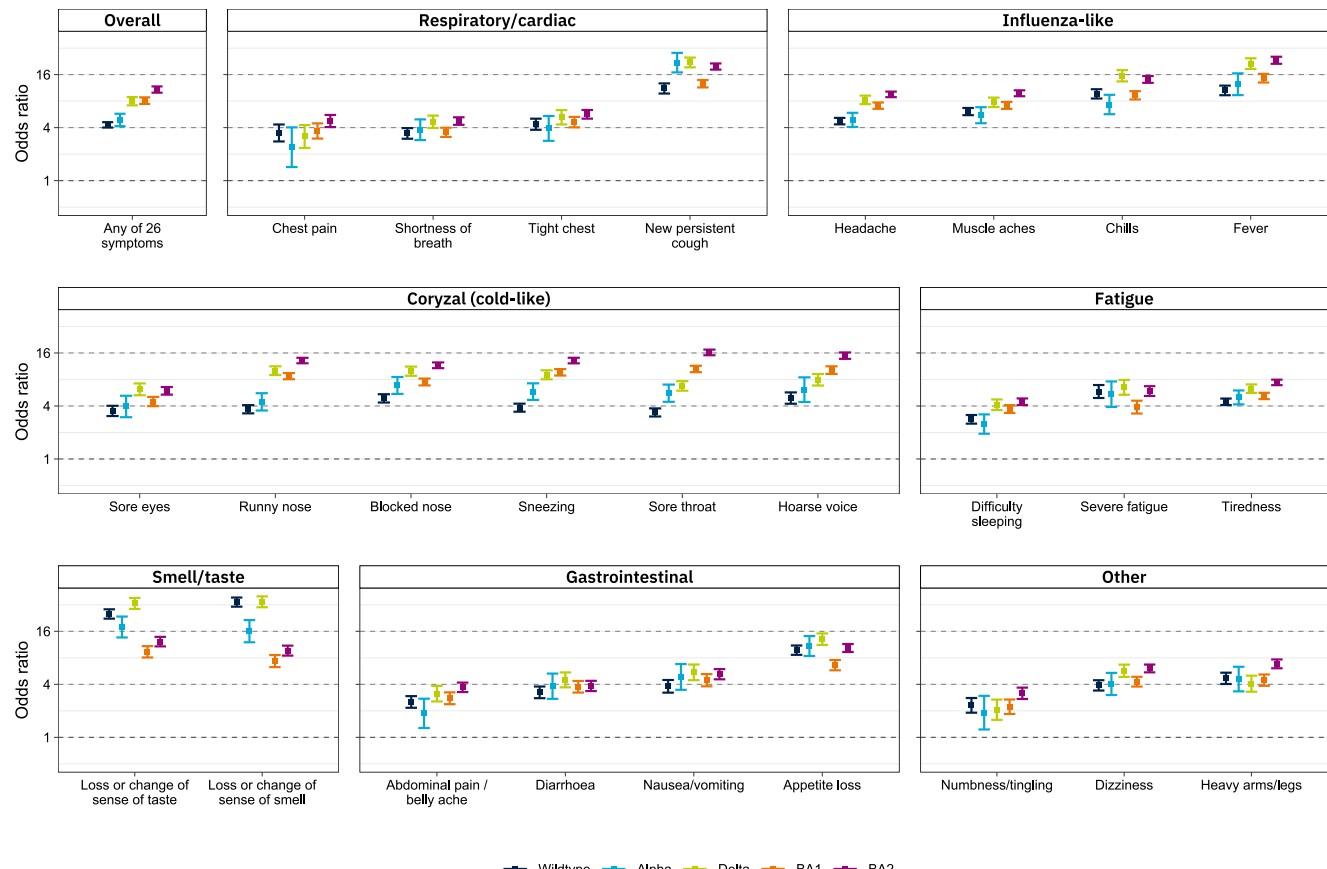

**Fig. 2 | Comparison of ORs for swab positivity based on presence or absence of any of 26 symptoms surveyed in N = 1,542,510 participants across five variant-phases of REACT-1.** ORs are derived from logistic regression models with swab positive (1/0) as the outcome variable, adjusted for age, sex and vaccination status.

Error bars show 95% confidence intervals. ORs are higher for BA.2 than BA.1 for all symptoms. Fever and cough have the highest ORs for BA.2 and BA.1, while loss or change of smell or taste have the highest ORs in all previous variants.

## Multivariable analysis for variable selection

We used Least Absolute Shrinkage and Selection Operator (LASSO) penalised logistic regression to identify parsimonious symptom sets selected as jointly and positively predictive of swab positivity for each variant (Fig. 3, Fig. S3); this method takes into account differences in symptom cooccurrence by variant (Figs. S4 and S5). Loss or change of sense of taste, new persistent cough, and fever were selected for each variant. Notably, cold-like symptoms of runny nose, sore throat, sneezing and hoarse voice were only selected for Omicron (BA.1 and BA.2).

## Omicron (BA.1 and BA.2)

Comparing symptoms for BA.2 vs BA.1 using logistic regression (based on either model adjustment or matching see "Methods"), infection with BA.2 was positively associated with chest pain, severe fatigue, runny nose, muscle aches, sneezing, fever, chills, tiredness, blocked nose and headache (in both sets of analyses); in unmatched analyses, infection with BA.2 was further associated with sore eyes, appetite loss and new persistent cough (Fig. 4).

In a subgroup of 5,598 double- and triple-vaccinated swab-positive individuals with BA.1 or BA.2, those infected with BA.2 were 54% more likely to report symptoms that interfered with their ability to carry out day-to-day activities 'a lot' (OR 1.54 [1.16, 2.06]) vs 'a little', 'not at all', or not reporting any symptoms, after adjustment for age group, sex, vaccine count, time since most recent vaccine, prior SARS-CoV-2 infection, time since symptom onset and calendar time (Table 1). In the same models, men were 38% less likely than women to report symptoms that interfered with their ability to carry out day-to-day

activities 'a lot' (0.62 [0.52,0.73]). Vaccine booster status and time since vaccination were not associated with ability to carry out daily activities. In the same subgroup, a log-linear regression of symptom count found that those infected with BA.2 reported 14% more symptoms, on average, than those with BA.1 (OR = 1.14 [1.10,1.19]) after adjustment for the same covariates as above (Supplementary Table 5).

**Ct values.** Ct values were lower for BA.2 than BA.1 (Supplementary Fig. 1). This may reflect the timing of the sampling with respect to the growth of the variant since more recent infections will tend to have lower Ct values (see Supplementary Table 2 and Supplementary Figs. 6 and 7, which show a positive correlation between time-since-symptom-onset and Ct values, and that mean time since symptom onset was lower for BA.2 than for BA.1). As expected, symptomatic individuals had lower Ct values (higher viral loads) than asymptomatic people. In linear regression models among swab positive individuals in rounds 17–19 (5 January to 31 March 2022), for each of the 26 surveyed symptoms, symptom reporting was associated with a lower Ct value. The lowest adjusted Ct values were for influenza-like or cold-like symptoms: fever, chills, sore throat, muscle aches, runny nose, sneezing and headache (Fig. 5), which frequently co-occurred (Supplementary Fig. 4). With the exception of fever, these symptoms were also commonly reported as the first symptom among symptomatic swab positives (Supplementary Fig. 5, Supplementary Table 2).

## Discussion

In this study of more than 1.5 million adults randomly selected from the population in England, we show differences in symptom reporting

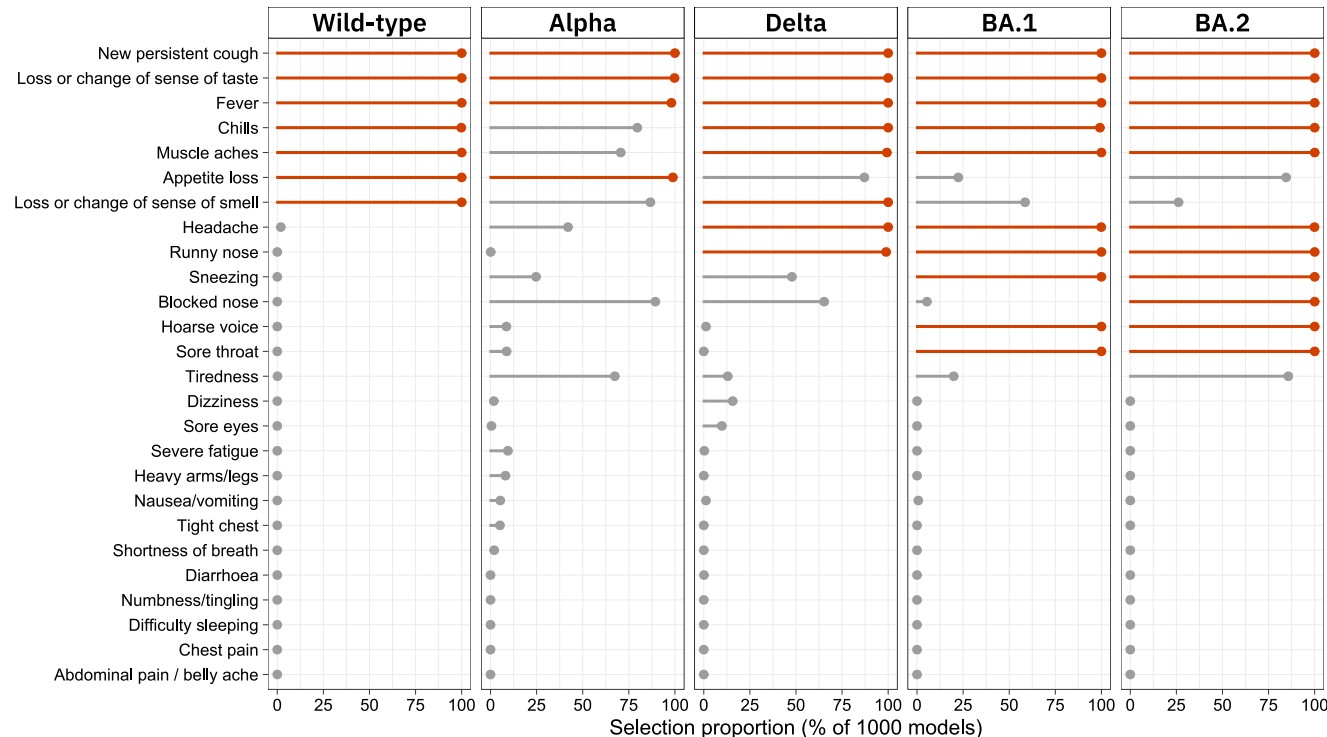

**Fig. 3 | Results of LASSO stability selection proportions with swab positive/ negative as the binary outcome variable and each of 26 symptoms as predictors, for five SARS-CoV-2 variants in England.** Age, sex and, where appropriate, vaccination status are forced into the models as unpenalised variables; regression coefficients for the symptoms are constrained to be positive. The selection proportion indicates the proportion of LASSO models, trained on subsamples of the data, in which each symptom was selected as a predictor.

associated with Omicron compared with previous variants, and within Omicron for BA.2 vs BA.1. This may reflect changes in the underlying pathophysiology associated with different variants, affecting, for example, receptor binding, cell entry or host response, against a background of differing levels of population immunity (both from natural infection and vaccine-induced)[13–15].

We found that loss or change of sense of smell or taste were less predictive of swab positivity for Omicron than for other variants, and that cold-like symptoms were more predictive for Omicron than for previous variants. Both these findings were consistent with previous reports[5,16,17]. Specifically, infections with Omicron variants are not as strongly associated with anosmia compared with previous variants. The loss of sense of smell or taste following infection with earlier variants of SARS-CoV-2 results from the downregulated expression of olfactory receptors[18]. It is possible that changes in the sequence of viral genes that regulate host responses in Omicron reduce this effect; detailed transcriptomic studies in animal models and humans may help to pinpoint the mechanisms involved.

Comparison of the intrinsic severity of SARS-CoV-2 variants is complex, owing to changing levels of population immunity due to prior infection or vaccination[15]. However, the rapid replacement of BA.1 by BA.2, and the large number of PCR positives, afforded an opportunity for comparison of the symptom burden and symptom severity of the two variants within a population with similar characteristics against a similar background of non-COVID-related illness and symptoms.

Comparing Omicron BA.2 with BA.1, we found that those with BA.2 were more likely to be symptomatic, to report a number of influenza-like and cold-like symptoms, and, in adjusted models, to report more symptoms, and to report that their symptoms affected their day-to-day activities 'a lot'. The last two findings were robust to adjustment for vaccine booster status and time since most recent

vaccine dose and are therefore unlikely to be explained by vaccination status or waning immunity following vaccination. The effects were somewhat attenuated by the addition of time since symptom onset and calendar time, suggesting that the higher symptom burden and severity of BA.2 (vs BA.1) may to some extent reflect the detection of swab positivity earlier in the disease course for BA.2; this is consistent with the higher transmissibility of BA.2 in a highly vaccinated population. Nonetheless, following adjustment, BA.2 was associated with 54% greater odds of symptoms affecting day-to-day activities 'a lot', and reporting of one additional symptom, on average, compared to BA.1.

While other studies of the BA.2 and BA.1 variants suggested that they were of similar severity[19,20] in terms of case hospitalisation rate or case fatality rate, the greater symptom burden and severity for BA.2 shown here may still be associated with substantial disruption to daily living, and have wider societal and economic impact.

From 1 April 2022 the UK government moved to a policy of 'living with COVID'[21]. With the lifting of restrictions and limited access to free testing limited, identifying individuals who are particularly likely to be infectious on the basis of symptoms alone may help reduce ongoing transmission of SARS-CoV-2. We show that in the Omicron period reporting fever, chills, sore throat, muscle aches, runny nose, sneezing and headache was associated with the lowest adjusted Ct values and therefore most likely to be indicative of higher viral load and increased infectiousness.

Our study has limitations. Response rates varied between 11.7% and 26.5% for rounds 2–19, so the samples may not be fully representative of, or results fully generalisable to, the population. Nevertheless, our random community sampling procedure included individuals from all of the 315 lower tier local authority areas in England in each round, ensuring wide geographical coverage and socioeconomic and demographic diversity. The symptoms surveyed were not exhaustive but, while not specific to COVID-19, were all shown to

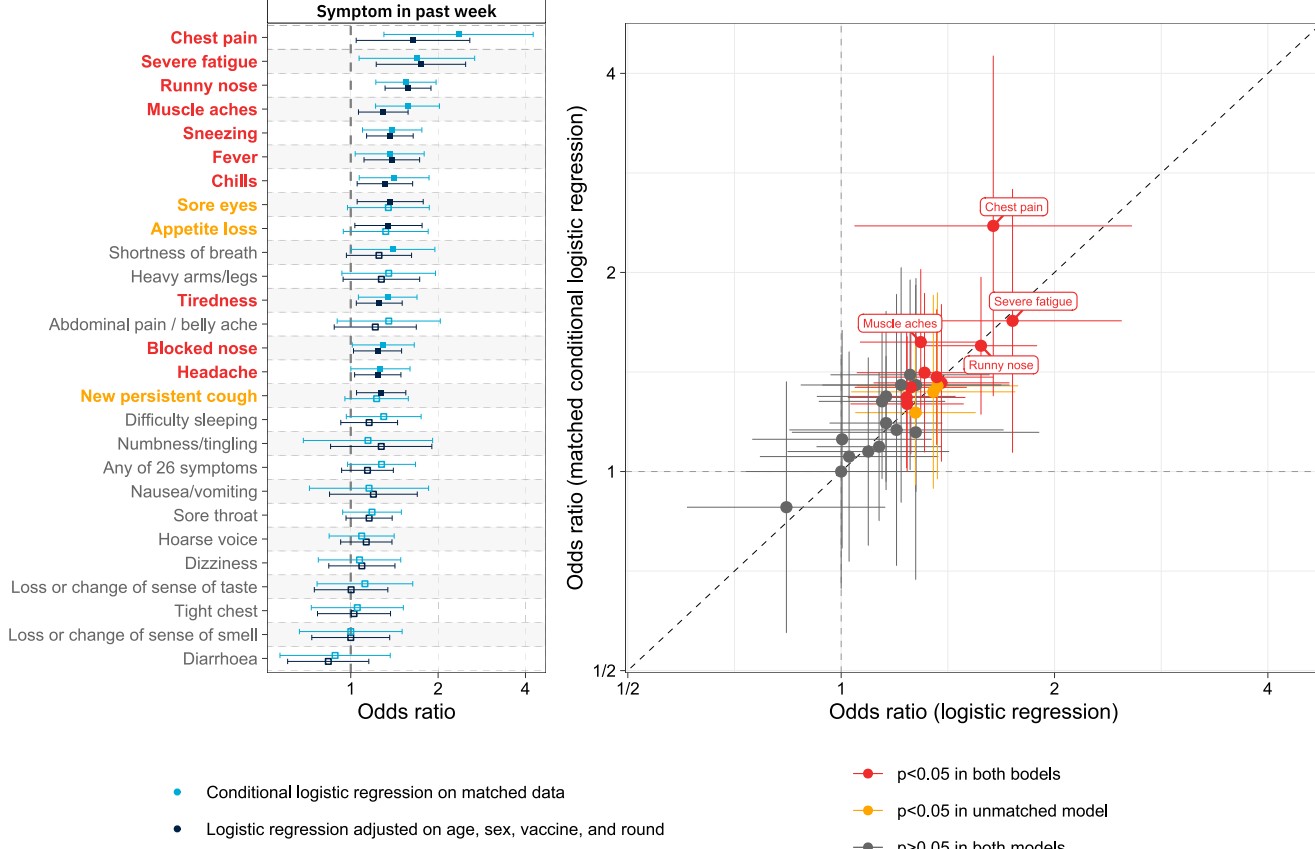

**Fig. 4 | ORs for infection with BA.2 vs BA.1 among swab-positive respondents.** ORs are derived from (i) logistic regression models with BA.2 vs BA.1 as the binary outcome variable, and presence or absence of any of 26 symptoms as explanatory variables, adjusted for age group, sex, round and vaccination status, among N = 5598 swab-positive individuals with either BA.2 or BA.1 in rounds 17–19 (5 January to 31 March 2022); and (ii) conditional logistic regression models with BA.2 vs BA.1 as the outcome variable among 1510 swab-positive individuals with either the BA.2 or BA.1 variant in rounds 17–19, matched 1:1 on age (±5 years), sex, vaccination status and round. In left panel, bars show 95% confidence intervals, and symptoms are ordered by mean OR across both models. Right panel directly plots the ORs from the two models for comparison. In both analyses, infection with BA.2 (vs BA.1) is positively associated with chest pain, severe fatigue, runny nose, muscle aches, sneezing, fever, chills, tiredness, blocked nose and headache; in unmatched analysis, infection with BA.2 is further associated with sore eyes, appetite loss and new persistent cough.

be predictive of SARS-CoV-2 swab positivity. Our analysis covers a period of 22 months, during which time background levels of natural and vaccine-acquired immunity varied substantially, making it difficult to differentiate the effect of viral mutations from the impact of vaccines and prior infection[15]. As REACT-1 data collection was non-continuous, we may have captured different stages of epidemic growth across variants, which may have differentially affected symptom reporting at different times.

Of those who provided valid swabs and consented to linkage in rounds 1–19 of REACT-1 (2,191,597 people in total), approximately 3% (65,915 people) participated in more than one round. On this basis, a correction factor of 1.015 could therefore be applied to the standard error estimates. We are not able to definitively identify instances of participation in more than one round among those who did not consent to linkage. However, because the consent-based estimate of the correction factor is so close to one, we feel confident reporting uncorrected standard errors and confidence intervals.

In summary, we have detected differences in symptom profiles reported during nearly 2 years of the COVID-19 epidemic in England, reflecting the emergence of different variants over that period against a background of varying immunity from prior infection and vaccination. Most recently, infection with Omicron is associated with lower reporting of loss or change of sense of smell and taste, and higher reporting of cold-like and influenza-like symptoms. Sequence-

confirmed BA.2 was associated with reporting of more symptoms and greater disruption to daily activity compared with BA.1. As routine testing becomes more limited in many countries, and as new variants emerge, understanding the symptom profiles which can identify individuals with a higher risk of transmission will become increasingly important.

## Methods

### Study population

The REACT-1 study has been tracking the prevalence of SARS-CoV-2 in the general population of England from 1 May 2020 to 31 March 2022. The study protocol and methodology have been published;[2,22] briefly, every 4–6 weeks, recruitment letters were sent to a random, nationally representative sample of people aged 5 years and over in England, using the National Health Service patient register. Participants then obtained self-administered throat and nasal swabs for SARS-CoV-2 PCR testing and completed an online or telephone questionnaire which included questions on demographic variables, behaviour, and recent symptoms. Questionnaires for each of the 19 completed rounds since May 2020 are available on the study website (https://www.imperial.ac.uk/medicine/research-and-impact/groups/react-study/for-researchers/react-1-study-materials/). Between 95,000 and 175,000 viable swabs and valid responses were gathered each round, with respondents unaware of their test result at the time of their response.

**Table 1 | Results from logistic regression of the response to the question "How much, if at all, do the symptoms you have had in the last 7 days reduce your/their ability to carry out day-to-day activities?" as a function of BA.2 / BA.1 infection, age group, sex, booster vaccine received (y/n), weeks since most recent vaccine, prior COVID-19 (28 days or more before testing), weeks since symptom onset, and calendar time (since 1 Jan 2021) among 5,637 double- or triple-vaccinated swab-positive individuals with either BA.2 or BA.1 infection**

| Variable | Category | Crude model | Plus age | Plus sex | Plus boosted | Plus weeks since vaccination | Plus prior COVID-19 | Plus weeks since symptom onset | Plus calendar time (weeks) |
|---|---|---|---|---|---|---|---|---|---|
| Omicron variant | BA.1 [ref] | – | – | – | – | – | – | – | – |
| | **BA.2** | **1.86 (1.59,2.18)** | **1.93 (1.64,2.28)** | **1.94 (1.64,2.29)** | **1.97 (1.66,2.33)** | **1.91 (1.58,2.29)** | **1.92 (1.59,2.30)** | **1.70 (1.40,2.06)** | **1.54 (1.16,2.06)** |
| Age | 18–24 [ref] | | – | – | – | – | – | – | – |
| | 25–34 | | 1.22 (0.80,1.84) | 1.26 (0.83,1.91) | 1.26 (0.83,1.92) | 1.26 (0.83,1.91) | 1.27 (0.83,1.93) | 1.13 (0.73,1.74) | 1.13 (0.74,1.75) |
| | 35–44 | | 1.64 (1.09,2.46) | 1.70 (1.14,2.56) | 1.73 (1.15,2.60) | 1.71 (1.14,2.57) | 1.72 (1.14,2.58) | 1.63 (1.07,2.49) | 1.65 (1.08,2.51) |
| | 45–54 | | 2.07 (1.39,3.10) | 2.19 (1.47,3.29) | 2.25 (1.49,3.38) | 2.21 (1.47,3.33) | 2.20 (1.46,3.32) | 2.09 (1.37,3.19) | 2.12 (1.39,3.25) |
| | 55–64 | | 2.12 (1.42,3.16) | 2.30 (1.54,3.43) | 2.36 (1.57,3.55) | 2.31 (1.54,3.48) | 2.32 (1.54,3.48) | 2.24 (1.47,3.41) | 2.28 (1.49,3.49) |
| | 65–74 | | 1.24 (0.81,1.88) | 1.39 (0.91,2.11) | 1.43 (0.93,2.19) | 1.39 (0.90,2.14) | 1.37 (0.89,2.11) | 1.33 (0.85,2.08) | 1.36 (0.86,2.13) |
| Sex | Female [ref] | | | – | – | – | – | – | – |
| | Male | | | 0.57 (0.48,0.67) | 0.57 (0.48,0.68) | 0.57 (0.48,0.68) | 0.57 (0.48,0.68) | 0.62 (0.52,0.73) | 0.62 (0.52,0.73) |
| Boosted (Yes) | No [ref] | | | | – | – | – | – | – |
| | Yes | | | | 0.90 (0.71,1.15) | 0.97 (0.72,1.31) | 0.96 (0.71,1.29) | 0.92 (0.67,1.27) | 0.88 (0.62,1.24) |
| Weeks since last vaccination | | | | | | 1.00 (0.99,1.01) | 1.00 (0.99,1.01) | 1.01 (0.99,1.02) | 1.00 (0.99,1.02) |
| Prior COVID-19 (28+ days ago) | No [ref] | | | | | | – | – | – |
| | Yes | | | | | | 0.64 (0.47,0.87) | 0.92 (0.65,1.30) | 0.92 (0.65,1.29) |
| Weeks since symptom onset | | | | | | | | 0.81 (0.66,0.98) | 0.81 (0.67,0.98) |
| Calendar time (weeks since 1 Jan) | | | | | | | | | 1.02 (0.98,1.05) |

Each column shows the addition of one covariate to the model. Odds ratios and 95% confidence intervals are shown. BA.2 infection (vs BA.1) is associated with increased risk of reduced ability to carry out day-to-day activities. This effect is robust to adjustment. Vaccine booster status was not associated with a change in ability to carry out daily activities.

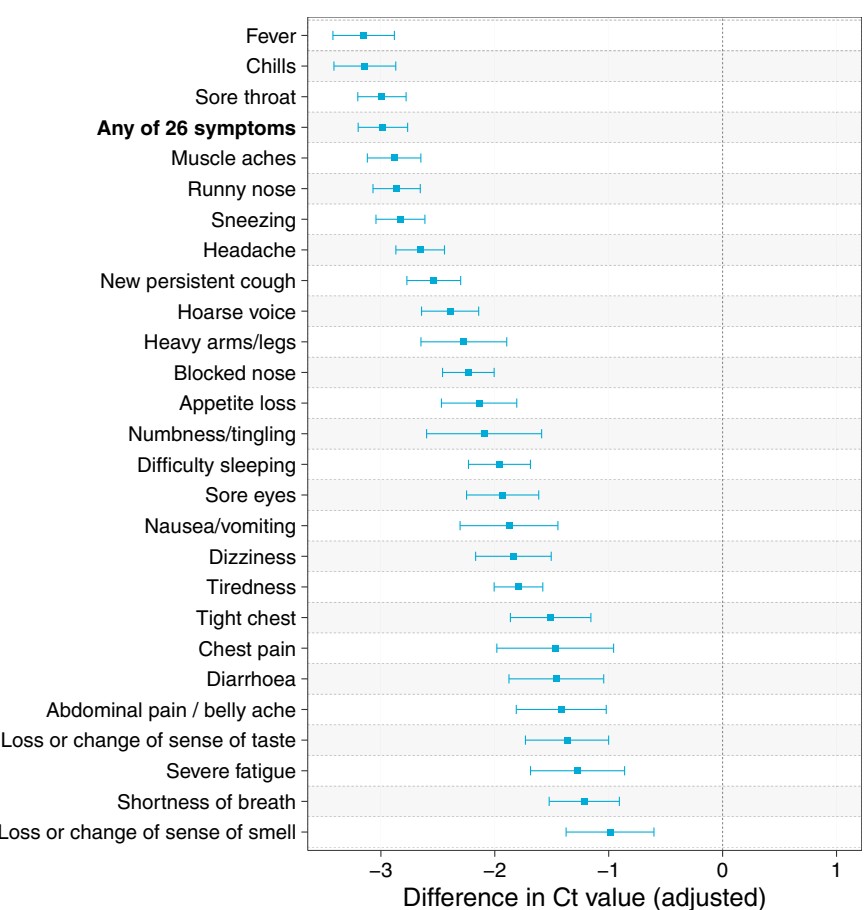

**Fig. 5 | Results of linear regression models with N-gene Ct values as the outcome variable and symptoms as individual predictors, adjusted for age, sex and, where appropriate, vaccination status, among N = 10,709 swab-positive respondents in rounds 17–19 (5 January to 31 March 2022).** Error bars show 95% confidence intervals. Fever, chills, and sore throat are the symptoms with the strongest negative association with Ct value, each associated with approximately a tenfold increase in viral load.

Participants were asked whether they experienced any of a list of 26 potential COVID-19 symptoms in the week prior to their test. These included loss or change of sense of smell or taste, respiratory/cardiac symptoms (new persistent cough, chest pain, tight chest, shortness of breath), cold-like symptoms (runny nose, blocked nose, sneezing, sore throat, hoarse voice, sore eyes), influenza-like symptoms (fever, chills, muscle aches, headache), gastrointestinal symptoms (nausea/vomiting, abdominal pain/belly ache, diarrhoea, appetite loss), fatigue-related symptoms (tiredness, severe fatigue, difficulty sleeping), and others (dizziness, heavy arms or legs, numbness/tingling).

We split data from 15 rounds of REACT-1 between 19 June 2020 and 31 March 2022 into distinct phases that correspond with the dominance of different SARS-CoV-2 variants in England: rounds 2–7 (at approximately monthly intervals between 19 June and 3 December 2020), when wild-type was dominant; rounds 8–10 (between 6 January and 29 March 2021), when Alpha (B.1.1.7) was dominant; rounds 13–15 (between 24 June and 5 November 2021), when Delta (B.1.617.2) was dominant; and rounds 17–19 (between 5 January and 31 March 2022), when Omicron (B.1.1.529) was dominant. In rounds 17–19 we use sequencing data to identify those participants who were infected with BA.1 or BA.2. Round 1 is excluded because the symptom questions asked were not consistent with subsequent rounds. Rounds 11, 12 and 16 are excluded from analysis because they occurred at times when two variants were competing for dominance in the population[23].

Adults aged 18 years and over were included in the analysis. A total of 266,847 participants were excluded because of missing symptom data (see supplementary methods for more details on data exclusions), and 38 were excluded because of missing age or sex data resulting in a final study population, after exclusions, of 1,542,510 participants.

**Statistical analyses**

We used univariable logistic regression models to estimate the risk of PCR swab-positivity for each variant conditional on experiencing each of the 26 symptoms. Models were adjusted for age group, sex, and self-reported vaccination status (coded as the number of vaccines received). Odds ratios and 95% confidence intervals are reported for each symptom and each variant. We also conducted a pooled analysis in which we tested the interactions between variants and each symptom in relation to PCR positivity, while additionally adjusting for calendar time, to assess the effect of differing SARS-CoV-2 prevalence and background symptom prevalence on the estimated odds ratios.

Variable selection models were trained on 70% of the data set, with 30% held back for model performance evaluation (see Supplementary Methods). We used stability selection applied to least absolute shrinkage and selection operator (LASSO) penalised logistic regression, with swab positivity as the binary outcome variable, and the 26 symptoms as predictors. To adjust for age, sex and vaccination status, these were included as unpenalised variables. The regression coefficients for selected symptoms were constrained to non-negative

values. LASSO models were fit on 1000 random 50% subsamples of the 70% training data. The proportion of models in which each symptom was selected is taken as a measure of variable importance. The threshold in selection proportion for final variable selection was calibrated in conjunction with the LASSO penalty parameter using an internal stability score[24].

**BA.2 vs BA.1.** Omicron BA.2 and BA.1 lineages were determined using viral genome sequencing on swab-positive swabs from rounds 17–19. We compared the symptom profiles among (BA.2 or BA.1) swab-positive individuals using logistic regression with BA.2 vs BA.1 as the binary outcome variable and each of the 26 symptoms as explanatory variables, adjusted on age group, sex, vaccination status and round. As a sensitivity analysis, we 1:1 matched swab-positive participants with BA.2 or BA.1 on age group (±5 years), sex, vaccination status and round in rounds 17–19, and conducted conditional logistic regression with BA.2 vs BA.1 as the binary outcome variable. We also used log-linear regression to compare symptom burden, in terms of number of symptoms experienced over the disease course, in double- and triple-vaccinated individuals with Omicron BA.2 and BA.1 (Supplementary Methods).

**Severity of symptoms.** To assess whether there are differences in symptom severity between BA.2 and BA.1 independent of vaccination history we took a subset of swab-positive individuals with sequence-confirmed BA.2 or BA.1 who had received second or third vaccines at least two weeks before their PCR test. In this group, we used logistic regression to compare the risk of reporting symptoms that affected their daily activities 'a lot' vs 'a little' or 'not at all' in people infected with BA.2 vs BA.1. We adjusted for age, sex, vaccine boosted (y/n), days since most recent vaccination, prior COVID-19 (28 days or more before test date), time since symptom onset, and calendar time (to account for seasonal effects). Odds ratios were reported for sequential models, with additional covariates added incrementally in the order described. We also used the same subset of individuals to model symptom count using multivariable log-linear regression models, again adding covariates sequentially and reporting odds ratios.

**Ct values.** Finally, we investigated the relationship between N-gene Ct value and symptom profile among swab positive individuals in rounds 17–19 (>95% Omicron), using linear regression models with Ct value as the outcome variable and each symptom separately as the independent variable. We also compared Ct values between swab-positive individuals with BA.2 or BA.1 using an unpaired Wilcoxon test, and compared Ct values in those excluded from analysis because of missing symptom data. Finally, we used linear regression to associate Ct values with time since symptom onset, in individuals with BA.2 or BA.1.

**Sensitivity analyses.** To assess possible biases introduced by non-continuous sampling, which might capture different stages of symptom onset for different variants, we (i) investigated the distributions of 'days since symptom onset' for different variants, (ii) investigated the relationship between symptom burden, in terms of number of symptoms reported, and days since symptom onset, and (iii) repeated the main analysis, further adjusting for time since symptom onset in the models.

**Data collection and software.** All data collection was captured with Questback (Sprint 2020 Installation). Data were analysed using R version 4.0.5[25].

**Reporting summary**
Further information on research design is available in the Nature Research Reporting Summary linked to this article.

## Data availability

The original datasets generated or analysed, or both, during this study are not publicly available because of governance restrictions and the identifiable nature of the data. Requests for access to raw data should be addressed to the corresponding authors and will be answered within 12 weeks. Summary statistics, descriptive tables, and code from the current REACT-1 study are available at https://github.com/mrc-ide/reactidd/tree/master/inst/extdata/variant_symptom_profiling_paper. REACT-1 study materials are available for each round at https://www.imperial.ac.uk/medicine/research-and-impact/groups/react-study/for-researchers/react-1-study-materials/ Sequence read data are available without restriction from the European Nucleotide Archive at https://www.ebi.ac.uk/ena/browser/view/PRJEB37886, and consensus genome sequences are available from the Global initiative on sharing all influenza data (GISAID)[26]. GISAID accession numbers for all sequences in the REACT1 study have been published in supplementary data file 1 of Eales et al.[27].

## Code availability

Scripts for this paper are available at https://github.com/mrc-ide/reactidd/tree/master/inst/extdata/variant_symptom_profiling_paper.

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

## Acknowledgements

The study was funded by the Department of Health and Social Care in England. The funders had no role in the design and conduct of the study; collection, management, analysis, and interpretation of the data; and preparation, review, or approval of this manuscript. MW is supported by grants from NIHR and UK Research and Innovation (UKRI): REACT GE (MR/V030841/1) and REACT Long COVID (REACT-LC) (COV-LT-0040). JE is an NIHR academic clinical fellow in infectious diseases. HW acknowledges support from an NIHR Senior Investigator Award, the Wellcome Trust (205456/Z/16/Z), and the NIHR Applied Research Collaboration (ARC) North West London. GC is supported by an NIHR Professorship. CAD acknowledges support from the MRC Centre for Global Infectious Disease Analysis, the NIHR Health Protection Research Unit in Emerging and Zoonotic Infections and the NIHR-funded Vaccine Efficacy Evaluation for Priority Emerging Diseases (PR-OD-1017-20007). MC-H and BB acknowledge support from Cancer Research UK, Population Research Committee Project grant 'Mechanomics' (grant No 22184 to MC-H). MC-H acknowledges support from the H2020-EXPANSE (Horizon 2020 grant No 874627) and H2020-LongITools (Horizon 2020 grant No 874739). PE is Director of the Medical Research Council (MRC) Centre for Environment and Health (MR/L01341X/1, MR/S019669/1). PE acknowledges support from Health Data Research UK (HDR UK); the National Institute for Health Research (NIHR) Imperial Biomedical Research Centre; NIHR Health Protection Research Units in Chemical and Radiation Threats and Hazards, and Environmental Exposures and Health; the British Heart Foundation Centre for Research Excellence at Imperial College London (RE/18/4/34215); and the UK Dementia Research Institute at Imperial College London (MC_PC_17114).

## Author contributions

M.W.: conceptualisation, methodology, formal analysis, investigation, data curation, writing—original draft, writing—review and editing, visualisation; J.E.: conceptualisation, methodology, investigation, writing—original draft, writing—review and editing; B.B.: conceptualisation, methodology, software, writing—original draft; W.B.: conceptualisation, methodology, investigation, writing—original draft; H.W.: conceptualisation, methodology, writing—original draft, writing—review and editing, supervision, funding acquisition; G.C.: conceptualisation, methodology, writing—original draft, writing—review and editing, investigation, supervision, funding acquisition; C.A.D.: conceptualisation, methodology, writing—original draft, writing—review and editing, investigation, supervision; M.C.-H.: conceptualisation, methodology, investigation, writing—original draft, writing—review and editing, supervision; P.E.: conceptualisation, methodology, investigation, writing—original draft, writing—review and editing, supervision, funding acquisition.

## Ethics

We obtained research ethics approval from the South Central-Berkshire B Research Ethics Committee (IRAS ID: 283787). Notification of favourable opinion and brief summary of the protocol are available here: https://www.hra.nhs.uk/planning-and-improving-research/application-summaries/research-summaries/react1-covid-19-uph/. Participants provided informed consent for their data to be used and, separately, indicated whether they were willing for their data to be linked to their NHS records.

## Public involvement

A Public Advisory Panel provides input into the design, conduct, and dissemination of the REACT research program.

## Competing interests

The authors declare no competing interests.
