## [Peer Review File · Nature Communications]

Variant-specific symptoms of COVID-19 in a study of 1,542,510 adults in EnglandREVIEWER COMMENTS

Reviewer #1 (Remarks to the Author):

The authors described the changing symptom profiles of multiple waves of COVID-19 in England, using the unique data from the REACT study. I agree with the authors that monitoring the changes in symptom profiles is getting more important in the transition from pandemic to endemic. With routine testing limited in many countries now, early treatment with antivirals might rely more on symptom monitoring. Please see below for my specific comments:

Major comments:

- 1) Although the reporting rate should be relatively stable in the REACT setting during different waves, it is not clear to me if there is any adjustment of the varying reporting behaviour over different waves. Given the high transmissibility of BA.2, is it possible that the REACT study "captured" the more severe infections in the severity pyramid in the BA.2 wave and therefore more symptoms were reported?
- 2) Preliminary findings from other studies about vaccines showed that vaccination is associated with less severe disease with fewer symptoms. Since the BA.2 wave came after the BA.1 wave in England, have effects of vaccination or waning of vaccine effectiveness been considered in the analysis?
- 3) Although I understand that disease severity is not necessarily associated with the number of reported symptoms, how should we interpret the findings that more symptoms were reported after BA.2 infections compared with wild-type and other VOCs, but in general BA.2 infection is considered "milder" compared with wild-type, Alpha, and Delta infections?
- 4) In Table S2, the BA.2 positives seem to be "younger" than BA.1. What's the role of age in the regression models? The authors described that LASSO logistic regression model have been used, but it is not clear what the final model looks like. Please consider adding more details of the variable selection in the appendix, e.g., the list of variables in the finalised models and the estimates of coefficients etc.

Minor comments:

- 1) Page 4, it should be "generalisable" but not "generalisible".

Reviewer #2 (Remarks to the Author):

This paper describes the symptom profiles of the main variants of SARS-CoV-2 and analyse the predictive strength of symptoms on positivity and viral load. The paper tackles an important question, relies on some high-quality data and his well-executed overall.

My main methodological concern is that the comparison of the predictive strength of symptoms between variants does not account for the underlying prevalence of COVID-19 and other respiratory viruses. If the prevalence of COVID-19 is higher (relative to other viruses), I would expect the symptoms to be stronger predictors of infection. If the virus is not widely circulating, then most people with symptoms are likely to have been infected with other viruses (or suffer from other conditions). As expected, the odds ratio for swab positivity of 'any of 26 symptoms' is highest for the most infectious variant BA.2. Similarly, the background prevalence of other respiratory viruses is likely to affect the ability to compare the predictive strength of the symptoms across different variants.

One way to account for the background prevalence would be to pool the data form all the rounds, and fit a model on the pooled dataset testing for the interaction between symptoms and variant type. You could then assess the effect of adjusting for background prevalence (and perhaps calendar month to capture seasonal effect) on the estimated ORs.

The approach used to compare symptoms between BA1 an BA2 is solid. However, it could be good

to test for the robustness of the results to an adjustment for calendar time (at time of interview/swab).

Minor comments:

1. In the Results section, the authors write that they 'included a total of 17,448 swab positive individuals' but in Table S2 the corresponding number seems to be 13,134 (It is entirely possible that I am misinterpreting this though!)
2. '266,847 participants were excluded because of missing symptom data' Could be good to know why they had missing symptoms? If someone has no symptoms, would they be likely to skip the question altogether. It could be good to see if they substantially differ from participants with valid symptoms data – mean CT value would be useful for instance

Finally, I would like to thank the authors for the opportunity to read a very interesting paper.

Reviewer response

REVIEWER COMMENTS

Reviewer #1 (Remarks to the Author):

The authors described the changing symptom profiles of multiple waves of COVID-19 in England, using the unique data from the REACT study. I agree with the authors that monitoring the changes in symptom profiles is getting more important in the transition from pandemic to endemic. With routine testing limited in many countries now, early treatment with antivirals might rely more on symptom monitoring. Please see below for my specific comments:

Major comments:

1) Although the reporting rate should be relatively stable in the REACT setting during different waves, it is not clear to me if there is any adjustment of the varying reporting behaviour over different waves. Given the high transmissibility of BA.2, is it possible that the REACT study “captured” the more severe infections in the severity pyramid in the BA.2 wave and therefore more symptoms were reported?

We recognise that response rates changed over time and that the non-continuous nature of the REACT-1 sampling may capture different stages of epidemic growth for the different variants.

We have added some additional analyses to interrogate this:

1. Table 1 and supplementary table 4 extend the severity analysis and present results from an additional number-of-symptoms analysis, adjusted for time since symptom onset. While the odds ratios/beta coefficients are attenuated by adjustment for calendar time and time since symptom onset, they robustly show excess odds of negative impact on daily activities, and higher symptom count, for BA.2 compared to BA.1.

2. Supplementary figure 7 shows three additional analyses of time since symptom onset. A and B show that time-since-symptom-onset was lower in the Omicron waves (and lower in BA.2 than BA.1), suggesting that we were sampling people earlier in the disease course, on average. C shows the main analysis further adjusted for time-since-symptom-onset. The odds ratios are attenuated but the relative magnitudes are consistent with the primary analysis shown in Figure 2.

These additional analyses are now briefly described in the Methods and Results and commented on in the Discussion.

2) Preliminary findings from other studies about vaccines showed that vaccination is associated with less severe disease with fewer symptoms. Since the BA.2 wave came after the BA.1 wave in England, have effects of vaccination or waning of vaccine effectiveness been considered in the analysis?

Thank you for suggesting this useful additional analysis.

We have added two analyses directly to address this question:

1. Table 1 (the logistic modelling of symptom severity), now includes double-vaccinated as well as triple-vaccinated (boosted) individuals and shows the odds ratios for triple vaccination (vs double), as well as time-since-most recent vaccination. Neither is significant at $p < 0.05$.
2. Supplementary Table 4 is a new analysis using the same population as in Table 1, with symptom count as the outcome and reporting beta coefficients. As with the first analysis, neither vaccination status nor time-since-vaccination is significant.

We have amended the Methods and Results accordingly and commented on this finding in the Discussion.

3) Although I understand that disease severity is not necessarily associated with the number of reported symptoms, how should we interpret the findings that more symptoms were reported after BA.2 infections compared with wild-type and other VOCs, but in general BA.2 infection is

considered “milder” compared with wild-type, Alpha, and Delta infections?

This is an important point. We have added two sentences to address this in the Discussion [para 5, Discussion].

4) In Table S2, the BA.2 positives seem to be “younger” than BA.1. What’s the role of age in the regression models? The authors described that LASSO logistic regression model have been used, but it is not clear what the final model looks like. Please consider adding more details of the variable selection in the appendix, e.g., the list of variables in the finalised models and the estimates of coefficients etc.

We have extended the Supplementary Methods section to include explicit specifications for each of the models. Supplementary Figure 3 shows the coefficients for each symptom in each of the stability runs, as well as the selected variables ranked by selection proportion and average coefficient magnitude. The boxplots show the median coefficient value for each symptom.

Minor comments:

1) Page 4, it should be “generalisable” but not “generalisible”.

Thank you, edited.

Reviewer #2 (Remarks to the Author):

This paper describes the symptom profiles of the main variants of SARS-CoV-2 and analyse the predictive strength of symptoms on positivity and viral load. The paper tackles an important question, relies on some high-quality data and his well-executed overall.

My main methodological concern is that the comparison of the predictive strength of symptoms between variants does not account for the underlying prevalence of COVID-19 and other respiratory viruses. If the prevalence of COVID-19 is higher (relative to other viruses), I would expect the symptoms to be stronger predictors of infection. If the virus is not widely circulating, then most people with symptoms are likely to have been infected with other viruses (or suffer

from other conditions). As expected, the odds ratio for swab positivity of ‘any of 26 symptoms’ is highest for the most infectious variant BA.2. Similarly, the background prevalence of other respiratory viruses is likely to affect the ability to compare the predictive strength of the symptoms across different variants.

One way to account for the background prevalence would be to pool the data from all the rounds, and fit a model on the pooled dataset testing for the interaction between symptoms and variant type. You could then assess the effect of adjusting for background prevalence (and perhaps calendar month to capture seasonal effect) on the estimated ORs.

Thank you – this is a good suggestion and we have added this pooled analysis to the paper. It is covered in the Methods and Results and the main findings are shown in Supplementary Figure 2. Our matched analysis from overlapping time periods to compare symptoms between BA.1 vs BA.2 (results in Figure 5) produces results consistent with the pooled analysis.

We have also added a plot (Supplementary Figure 9) showing the prevalence of any-of-26 symptoms in PCR positive individuals compared with PCR negative (ie background symptom prevalence).

The approach used to compare symptoms between BA1 and BA2 is solid. However, it could be good to test for the robustness of the results to an adjustment for calendar time (at time of interview/swab).

We have extended the analysis to show the effects of incrementally adding covariates, including calendar time (Table 1). We have also added another separate analysis of symptom count among the same population, again adjusting for calendar time (Supplementary Table S4). In both cases, the OR/betas for BA.2 (vs BA.1) are attenuated, but remain significant ($p < 0.01$).

Minor comments:

1. In the Results section, the authors write that they ‘included a total of 17,448 swab positive individuals’ but in Table S2 the corresponding number seems to be 13,134 (It is entirely possible

that I am misinterpreting this though!)

The 17,448 figure includes 4,314 individuals from rounds 17–19 who did not have their samples sequenced and were therefore excluded from the analysis of Omicron BA.1 and BA.2. We included them in the main table (supplementary table 1) and descriptive results to show the overall population characteristics and prevalence by survey phase.

We have added a note to supplementary table 2 to clarify this.

2. ‘266,847 participants were excluded because of missing symptom data’ Could be good to know why they had missing symptoms? If someone has no symptoms, would they be likely to skip the question altogether. It could be good to see if they substantially differ from participants with valid symptoms data – mean CT value would be useful for instance

The participants excluded because of missing symptoms were people who either skipped the top-level question ‘have you felt unwell in the past month?’ (the large majority) or said that they had said that they had felt unwell but then did not tick any of the specific symptoms later in the survey, or declare ‘none of these’ (486 people).

We have (i) added the relevant symptom questions to the supplementary material for clarity, (ii) added a data exclusions paragraph to the supplementary material, and (iii) added a plot comparing Ct values, PCR positivity rates and population size for those with missing symptoms vs other participants (Supplementary Figure 8 A and B). Those with missing symptom status are shown to have lower Ct values and higher PCR positivity than asymptomatics, but higher Ct values and lower PCR positivity than those with ‘classic’ COVID-19 symptoms (cough, fever, loss or change of smell or taste) or any other symptoms. This suggests that, as expected, the ‘unknown’ symptom status group contains a mixture of people with symptoms and without.

REVIEWERS' COMMENTS

Reviewer #1 (Remarks to the Author):

All my comments have been satisfactorily addressed. I don't have other comments.

Reviewer #2 (Remarks to the Author):

The authors have addressed most of my comments. However, Supplementary Figure 2 leaves me a bit unsatisfied. Couldn't you use the results from that model to replicate the ORs from Figure 2? I would fit

$y[\text{PCR pos/neg}] \sim \text{covariates} [\text{age, sex, vaccination}] + \text{variant} [\text{delta, BA1, BA2}] + \text{symptom} [y/n]$
+ variant:symptom [interaction]+ calendar time (which seems to be missing from your equation?)

That would help assess how sensitive the results from Figure 2 are from the background prevalence.

It is entirely possible that I am missing something obvious though!

Variant-specific symptoms of COVID-19 in a study of 1,542,510 adults in England: response to reviewers

REVIEWERS' COMMENTS

Reviewer #1 (Remarks to the Author):

All my comments have been satisfactorily addressed. I don't have other comments.

Reviewer #2 (Remarks to the Author):

The authors have addressed most of my comments. However, Supplementary Figure 2 leaves me a bit unsatisfied. Couldn't you use the results from that model to replicate the ORs from Figure 2? I would fit

$y[\text{PCR pos/neg}] \sim \text{covariates} [\text{age, sex, vaccination}] + \text{variant} [\text{delta, BA1, BA2}] + \text{symptom} [y/n] + \text{variant:symptom} [\text{interaction}] + \text{calendar time}$ (which seems to be missing from your equation?)

That would help assess how sensitive the results from Figure 2 are from the background prevalence.

It is entirely possible that I am missing something obvious though!

Thank you for this clarification. We have now amended the pooled analysis using the suggested specification, and report odds ratios as suggested. Supplementary figure 2, and the accompanying legend, have been updated, as well as a minor amend to the methods for clarity. The odds ratios are consistent with the primary analysis in figure 2.